# A Real-Time Trajectory Optimization Method for Hypersonic Vehicles Based on a Deep Neural Network

Jianying Wang [1], Yuanpei Wu [1], Ming Liu [2], Ming Yang [2] and Haizhao Liang [1,*]

1 School of Aeronautics and Astronautics, Sun Yat-sen University, Shenzhen 510275, China; wangjiany@mail.sysu.edu.cn (J.W.); wuyp33@mail2.sysu.edu.cn (Y.W.)
2 Science and Technology on Space Physics Laboratory, Beijing 100190, China; liumingbuaa@163.com (M.L.); yangmhitsat@126.com (M.Y.)
* Correspondence: lianghch5@mail.sysu.edu.cn

**Abstract:** Considering the high-efficient trajectory planning requirements for hypersonic vehicles, this paper proposes a real-time trajectory optimization method based on a deep neural network. First, the trajectory optimization model of the hypersonic vehicle reentry phase is developed. The pseudo-spectral method is used to perform the trajectory optimization offline, and multiple optimal trajectory data are obtained. In addition, based on the inherent relationship between the state and control variables of a trajectory, a neural network is established to predict the current control outputs. The sample library of optimal trajectory data is used to train the parameters of the deep neural network to obtain an optimal neural network model. Finally, the simulation verification of the hypersonic vehicle reentry phase is performed. The simulation results show that under the condition of the initial value deviation and environmental interference, the proposed deep learning-based method can achieve a fast generation of hypersonic vehicle optimal trajectories, while achieving the advantages of high computational efficiency and reliability. Compared to traditional trajectory optimization algorithms, the proposed method has the generalization capability that satisfies the accuracy requirements and meets the demands of online real-time trajectory optimization.

**Keywords:** hypersonic vehicle; pseudo-spectral method; trajectory optimization; deep learning; reentry phase

## 1. Introduction

In recent years, hypersonic vehicles have become one of the development directions in the aerospace field. A hypersonic vehicle is a vehicle that moves through the atmosphere at a height of below 90 km at a speed of above Mach 5. Under extreme and variable flight conditions, such as nonlinear aerodynamic parameters and high heat load, the dynamical system of a hypersonic vehicle is uncertain, coupled, and highly nonlinear. Accordingly, how to manipulate and control a hypersonic vehicle to meet particular requirements denotes a highly constrained nonlinear optimization problem.

In general, trajectory optimization of a hypersonic vehicle represents a process of designing a trajectory that minimizes (or maximizes) certain performance measures, while satisfying a set of constraints. Many numerical methods have been proposed to transform the continuous-time optimal control problem into an approximate, finite space, and precision range optimization problem in a certain way. Typically, there are two types of traditional methods to solve the optimal control problem: indirect methods and direct methods [1]. The indirect methods transform the optimal control problem into a Hamilton Boundary Value Problem (HBVP) using the Pontryagin minimum principle, and an optimal numerical solution of a trajectory can be obtained by solving the boundary value problem. Indirect methods have been used for solving hypersonic vehicle trajectory planning problems, which could provide a high accuracy solution [2–5]. However, due to the well-known drawbacks of complex implementation and high sensitivity to the initial condition

of the indirect methods, direct methods have been widely used since they do not require optimal necessary conditions. Namely, the direct methods discretize and parameterize the continuous optimal control problem and use numerical methods to find the optimal performance index [6]. Several popular direct methods, including the collocation method [7], and the pseudo-spectral method [8–11], have been extensively used for solving a variety of trajectory optimization problems. The direct methods have the advantages of a robust convergence domain and flexible applicability to practical complex problems. However, dealing with transformed numerical equations on each of the collocation points introduces much computation load, which cannot meet the computational efficiency requirements of online trajectory generation applications.

Due to the increasingly high demand on real-time engineering, how to provide a significant improvement in the algorithm calculation speed has become a challenge. Many studies have focused on exploration and improvement in real-time trajectory optimization based on the existing numerical methods. Antony [12] developed a graphical processing unit accelerated indirect ballistic optimization method using the multiple shot method and the extended method, which can maximize the computational efficiency, while taking full advantage of the parallelism characteristic of the indirect targeting method. To improve the computational efficiency of the Chebyshev pseudo-spectral method, Wang [13] used the differential flatness theory to solve the trajectory problem of hypersonic vehicles by reducing kinetic differential constraints, and the results showed that the solution time of a single trajectory was effectively reduced, compared with the traditional pseudo-spectral methods. In recent years, convex optimization techniques have attracted great attention due to their advantages of efficient solution and convergence property [14–20]. Wang [21] proposed two improved algorithms for the hypersonic vehicle's reentry trajectory optimization, named the line search sequence convex optimization and the trust domain sequence convex optimization, using the predictive correction method to find the initial 3D trajectory, which improves the convergence of the solution process. In addition, a robust trajectory optimization method combining chaotic polynomials and convex optimization techniques was proposed in [22,23]. This method exploits the high accuracy of chaotic polynomial algorithms for solving highly nonlinear dynamics problems and the high efficiency of convex optimization algorithms for solving optimal control problems. However, the convexification of the trajectory planning problem is still a challenge, especially for systems with high nonlinear dynamics and constraints. As mentioned above, most studies have improved the algorithm solution efficiency through mathematical processing using convex optimization methods, pseudo-spectral methods, or indirect methods. The principle of the improved algorithms still relies on the iterative convergence framework, where the selection of the iterative initial conditions directly affects the algorithms' convergence. Moreover, these solutions limit the online application of the algorithm to a certain extent.

Recently, taking the advantages of good generalization ability and rapidity, many mature machine learning methods have been proposed to achieve onboard application in order to meet the requirements for high autonomy, required optimality, and real-time performance [24–26]. Yin [27] proposed a DNN- (Deep Neural Network) based method for low-thrust orbit transfers, where the fast generation of optimal trajectories was achieved by the advantages of high computational efficiency and reliability. For the online trajectory planning for moon landings, Furfaro [28] proposed a deep convolutional neural network model to predict fuel-optimal control actions, using raw images taken by onboard optical cameras. Shi [29] proposed a deep learning-based approach for real-time trajectory optimization of hypersonic vehicles, and the trained DNN-based trajectory was demonstrated to be capable of generating optimal control commands onboard, while achieving good real-time performance and stable convergence. However, only a 2D flight dynamics model was considered, but it cannot fully describe 3D trajectories of hypersonic vehicles. Moreover, the terminal states of the trajectory planning problem were set as certain values, where the uncertainties of terminal states in different flight missions were ignored.

In this study, following the success of the machined learning method in the fast generation of optimal controls, a real-time DNN-based method is proposed to solve the optimal trajectory generation problem of a three-DOF (Degrees of Freedom) hypersonic vehicle reentry model. The proposed method has the generalization capability that satisfies the accuracy requirements and meets the demands of online real-time trajectory optimization better than the traditional trajectory optimization.

The contribution of this work is threefold. First, a DNN-based optimal control method that has the potential to address the long-standing challenge of solving highly nonlinear trajectory optimization problems for hypersonic vehicles, while achieving good real-time performance is proposed. Second, the pseudo-spectral method is used to generate optimal trajectories for network training efficiently. Third, extensive simulation results are provided to validate the performance of different DNN-based models in learning the nonlinear relationship to solve the trajectory optimization problem, and the accuracy of the trained DNN models is verified through the comparison with the direct approaches. The reference [29] proposed a real-time trajectory optimization method for hypersonic vehicles based on DNN models, which is potentially capable of near-optimal control with real-time performance and stable convergence. However, the proposed method only focused on the 2D (two-dimensional) trajectory optimization problem, and the trajectory end point was set to be fixed. The method proposed in [29] is limited to 3D trajectory optimization with random endpoint cases. To solve the problem, this paper proposed the 3D real-time trajectory optimization method based on the pseudo-spectral method and the DNN models, where the pseudo-spectral method is used to generate large-scale 3D optimal trajectory training data, and DNN models are designed and trained to predict optimal actions according to the flight states.

The remaining paper is organized as follows. Section 2 presents a continuous-time optimal control problem of a three-dimensional (3D) hypersonic flight, with nonlinear dynamics and terminal constraints, and introduces the research idea for solving the trajectory optimization problem of hypersonic vehicles. Section 3 describes the DNNs trained using the optimal trajectories obtained by the pseudo-spectral method. Section 4 provides the numerical simulation results to evaluate the performance of the proposed DNN-based trajectory optimization method. Section 5 concludes the paper and presents future work directions.

## 2. Materials and Methods

### 2.1. Three-DOF Dynamic Model Development

In this paper, the trajectory of a hypersonic vehicle is considered as a three-DOF reentry motion model of a rotating sphere, where the sideslip angle is zero. The position parameters, including the geocentric distance $r$, longitude $\theta$, and latitude $\varphi$, are defined in the geocentric spherical fixed coordinate system. The velocity parameters include the velocity $v$, track angle $\gamma$, and course angle $\psi$. The undynamic three-DOF reentry motion equations expressed by the above-listed parameters are as follows:

$$\frac{dr}{dt} = V \sin \gamma \tag{1}$$

$$\frac{d\theta}{dt} = \frac{V \cos \gamma \sin \psi}{r \cos \varphi} \tag{2}$$

$$\frac{d\varphi}{dt} = \frac{V \cos \gamma \cos \psi}{r} \tag{3}$$

$$\frac{dV}{dt} = -\frac{D}{m} - g \sin \gamma \tag{4}$$

$$\frac{d\gamma}{dt} = \frac{1}{V} \left[ \frac{L \cos \sigma}{m} + \left( \frac{V^2}{r} - g \right) \cos \gamma \right] \tag{5}$$

$$\frac{d\psi}{dt} = \frac{1}{V}\left(\frac{L \sin \sigma}{m \cos \gamma} + \frac{V^2}{r} \cos \gamma \sin \psi \, tan \, \varphi\right) \tag{6}$$

where the Earth rotation acceleration is assumed to be zero, and $g, \sigma, L, D$ represent the gravitational acceleration, roll angle, lift, and drag, respectively.

In order to improve the efficiency of the optimization process, a dimensionless method is applied to the undynamic three-DOF reentry model. The dimensionless geocentric distance $z$, velocity $u$, and flight time $\tau$ are, respectively, defined as:

$$z = r/R_0, u = \frac{V}{V_c}, \tau = t/\sqrt{R_0/g_0} \tag{7}$$

$R_0$ is the radius of the earth, and $g_0$ is the gravitational acceleration. The dimensionless three-DOF reentry equations can be obtained by substituting the above variables into Equations (1)–(6), which yields:

$$\frac{dz}{d\tau} = u \sin \gamma \tag{8}$$

$$\frac{d\theta}{d\tau} = \frac{u \cos \gamma \sin \psi}{z \cos \varphi} \tag{9}$$

$$\frac{d\varphi}{d\tau} = \frac{u \cos \gamma \cos \psi}{z} \tag{10}$$

$$\frac{du}{d\tau} = -\overline{D} - \frac{\sin \gamma}{z^2} \tag{11}$$

$$\frac{d\gamma}{d\tau} = \frac{1}{u}\left[\overline{L} \cos \sigma + \frac{\cos \gamma}{z}\left(u^2 - \frac{1}{z}\right)\right] \tag{12}$$

$$\frac{d\psi}{d\tau} = \frac{1}{u}\left[\frac{\overline{L} \sin \sigma}{\cos \gamma} + \frac{u^2}{z} \cos \gamma \sin \psi \, tan \, \varphi\right] \tag{13}$$

The dimensionless lift and drag are, respectively, defined as follows:

$$\overline{L} = \rho(uV_c)^2 S_{ref} C_L / (2mg_0) \tag{14}$$

$$\overline{D} = \rho(uV_c)^2 S_{ref} C_D / (2mg_0) \tag{15}$$

$\rho$, $m$, $S_{ref}$, $C_L$ and $C_D$ represent the air density, the mass, aerodynamic reference area, lift and drag coefficients of the aircraft, respectively, and $V_c = \sqrt{g_0 R_0}$. The control vector is expressed as $U = [\alpha, \sigma]$, which represents the generalized lift coefficient and heeling angle, respectively, and the fight trajectory can be generated after designing the changing curve of the control vector.

*2.2. Problem Statement*

The trajectory planning problem for a typical hypersonic vehicle is considered in this paper. It can be described as an optimization problem, the core of which is to choose optimal or suboptimal control parameters such that the objective function is minimized, while under constraints including boundary constraints, path constraints and constraints of control.

It is worth pointing out that the initial and final states in this research are considered random, which is more closer to the actual flight environment. Namely, the initial conditions $S_0 = [r_0, \theta_0, \varphi_0, V_0, \gamma_0, \psi_0]$, which represent initial geocentric distance, longitude, latitude, velocity, track angle and course angle, respectively, and the final conditions $S_f = [\theta_f, \varphi_f]$, which represent the terminal longitude and latitude, respectively, are given as random values within a certain range, and the solutions of the problem are proposed to gain the optimal or suboptimal trajectory based on the random cases.

In general, several types of performance indices to specify different optimization objectives exist, such as the maximum range, minimum heat load, and minimum time. In this paper, for the mission to reach the desired area fast, the total flight time is considered to be an important performance index, and the objective function is given by $min \ t_f$.

The process constraints mainly include the dynamic pressure constraint, heat flow constraint, and overload constraint. In view of the severe flight environment of a hypersonic vehicle, the following constraints need to be satisfied rigorously.

### 2.2.1. Dynamic Pressure Constraint

Dynamic pressure refers to the kinetic energy of a fluid per unit volume. In the field of hypersonic vehicles, the dynamic pressure is proportional to the aerodynamic force and torque. Considering the influence of the dynamic pressure on the requirement for lateral stability of the control system, the dynamic pressure in the reentry process needs to meet the following constraint:

$$q = \frac{1}{2}\rho V^2 \leq q_{max} \tag{16}$$

### 2.2.2. Heat Flow Constraint

Considering the stagnation point is an area where a vehicle is heated more severely, the heat flow of the stagnation is generally taken as a constraint. The heat flow constraint is given by:

$$\dot{Q} = K\left(\frac{\rho}{\rho_0}\right)^n \left(\frac{V}{V_c}\right)^m \tag{17}$$

### 2.2.3. Overload Constraint

The overload constraint needs to be considered in the reentry process for the purpose of structural safety. The overload constraint is defined as follows:

$$n = \sqrt{\overline{L}^2 + \overline{D}^2} = q\sqrt{C_D^2 + C_L^2}\frac{S}{mg} \leq n_{max} \tag{18}$$

### *2.3. Research Ideas*

In this paper, the DNN-based real time trajectory planning method is proposed. The whole process of the DNN-based real-time trajectory optimization is shown in Figure 1. First, the Chebyshev pseudo-spectral method is used to generate the optimal state–action samples [*x*,*a*]. In this way, the generation of large-scale optimal sample data, which is time consuming, is performed offline. Moreover, by normalizing and interpolating the discrete state and action data, the resulting optimal samples are obtained and sent to the neural network. Finally, the network is designed to learn the nonlinear functional relationship between the state and action. With the training process, the network that can output the optimal controls in accordance with the current flight state is derived. Based on the derived deep neural work, the trajectory planning and control can be performed online, since the calculation load of a network is quite acceptable as real-time output.

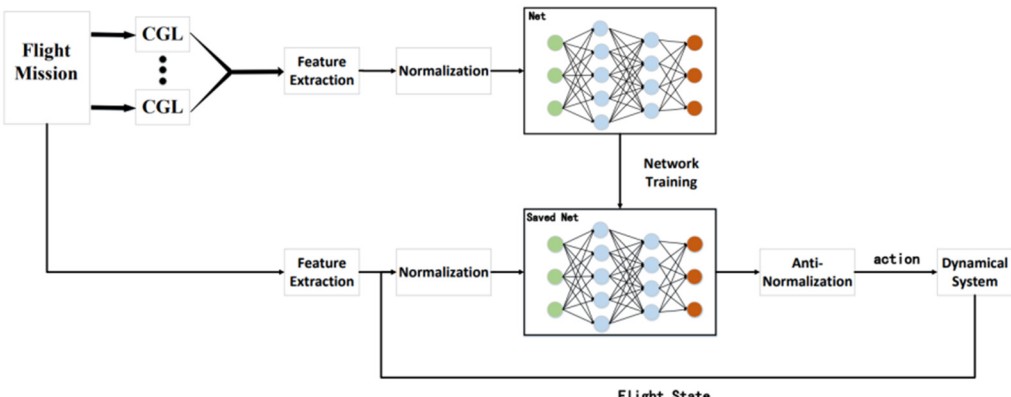

**Figure 1.** DNN-based real-time trajectory optimization.

### 3. Sample Data Generation Method Based on Chebyshev Pseudo-Spectral Method

*3.1. Chebyshev Pseudo-Spectral Method*

The basic solution steps of the Chebyshev pseudo-spectral method are as follows. Choose discrete continuous-time state and control variables over a series of CGL (Chebyshev–Gauss–Lobatto) points and construct the Lagrange interpolation polynomials using these discrete points as nodes to approximate the real state and control. Next, approximate the derivatives of the state variables over time by deriving global interpolated polynomials to convert differential equation constraints to algebraic constraints. Then, integrate the terms in the efficacy indicators, calculated by Clenshaw–Curtis numerical integration. Using the Chebyshev pseudo-spectral method, the optimal control problem can be transformed into an NLP (Nonlinear Programming) problem with a set of algebraic constraints.

Time-domain transformation:

The CGL points in the Chebyshev pseudo-spectral method are in the interval of $[-1, 1]$, so the time variable $t$ can be transformed to $\tau$ as follows:

$$\tau = \frac{2t}{t_f - t_0} - \frac{t_f + t_0}{t_f - t_0} \tag{19}$$

Calculation of discrete nodes:

In the Chebyshev pseudo-spectral method, discrete nodes are selected as extremal points of a Chebyshev polynomial of the $N$th order, i.e., the CGL points that are unevenly distributed in the range of $[-1, 1]$. For the standard CGL points, the definition of Legendre–Gauss point $\tau_k$ is as follows:

$$\tau_k = \cos\left(\frac{\pi k}{N}\right), \ k = 0, \ldots, N \tag{20}$$

Approximate interpolation of state and control variables:

The Lagrange interpolation polynomial is constructed as an approximation of the above state and control variables at $(N + 1)$ discrete points. The approximate expressions of the real state and control variables are, respectively, as follows:

$$\begin{aligned} \boldsymbol{x}(t) \approx \boldsymbol{x}^N(t) = \sum_{j=0}^{N} x_j \phi_j(t) \\ \boldsymbol{u}(t) \approx \boldsymbol{u}^N(t) = \sum_{j=0}^{N} u_j \phi_j(t) \end{aligned} \tag{21}$$

The Lagrange interpolation base function is defined as:

$$\phi_j(t) = \frac{(-1)^{j+1}}{N^2 c_j} \frac{(1 - t^2)\dot{T}_N(t)}{t - t_j} \tag{22}$$

In Equation (22), $c_j = \begin{cases} 2, j = 0, N \\ 1, 1 \leq j \leq N - 1 \end{cases}$, $t_j(j = 0, \cdots, N)$ represents the CGL points. Based on the nature of the Lagrange interpolation, the state approximation at a discrete node is equal to the actual state, while the control approximation is equal to the actual control.

Dynamic constraint processing:

Based on Equation (20), an approximate expression of the derivative of the state vector at time $t_k$ is given as:

$$\dot{x}(t_k) \approx \dot{x}^N(t_k) = \sum_{j=0}^{N} x_j \dot{\phi}_j(t_k) = \sum_{j=0}^{N} D_{kj} x_j \tag{23}$$

where $D_{kj}$ represents elements in a row $k$ and column $j$ of a $(N + 1) \times (N + 1)$ differential matrix $D$ that is expressed as:

$$D = \begin{cases} \frac{c_k}{c_j} \frac{(-1)^{k+j}}{t_k - t_j} & k \neq j \\ -\frac{t_k}{2(1-t_k^2)} & 1 \leq k = j \leq N - 1 \\ \frac{2N^2+1}{6} & k = j = 0 \\ -\frac{2N^2+1}{6} & k = j = N \end{cases} \tag{24}$$

The derivatives of the substituted state variables over time can be obtained by Equation (23) and discretized at the interpolation node. Thus, the kinetic differential equation constraints of the original optimal control problem can be converted to the algebraic constraints for $k = 0, 1, \cdots, N$ as follows:

$$\sum_{j=0}^{N} D_{kj} x(t_j) - \frac{\tau_f - \tau_0}{2} f(x(t_k), u(t_k), t_k) = 0 \tag{25}$$

where $f$ represents the state equation of the system. For the process constraints defined by the above equation, strict satisfaction at the discrete nodes is required.

Approximate integration of performance indicators:

When there is an integral term in the optimization performance metric, the Clenshaw–Curtis numerical integration can be used to approximate it. For a continuous function over the interval of $[-1, 1]$, its integration can be summed and approximated by the function at $(N + 1)$ discrete points of the CGL as follows:

$$\int_{-1}^{1} p(t)dt \approx \sum_{k=0}^{N} p(t_k)\omega_k \tag{26}$$

where $\omega_k(k = 0, 1, \cdots, N)$ denotes the Clenshaw–Curtis weight.

$$J \approx J^N = \Phi\left[\tilde{\zeta}(-1), \tilde{\zeta}(1), t_0, t_f\right] + \frac{t_0 - t_f}{2} \sum_{k=0}^{N} \omega_k^C g'(\tau_k) \Theta\left[\tilde{\zeta}(g(\tau_k))\right] \tag{27}$$

where $g'(\tau_k)$ is the first-order derivative of the conformal map, $\omega_k^C$ is the Clenshaw–Curtis weight, and it holds that:

$$for\ N\ is\ even, \begin{cases} \omega_0^C = \omega_N^C = \frac{1}{N^2-1} \\ \omega_s^C = \omega_{N-s}^C = \frac{4}{N} \sum_{i=0}^{\frac{N}{2}''} \frac{1}{1-4i^2} \cos\frac{2\pi is}{N}, \ s = 1, \ldots, \frac{N}{2} \end{cases} \\ for\ N\ is\ odd, \begin{cases} \omega_0^C = \omega_N^C = \frac{1}{N^2} \\ \omega_s^C = \omega_{N-s}^C = \frac{4}{N} \sum_{i=0}^{\frac{(N-1)}{2}''} \frac{1}{1-4i^2} \cos\frac{2\pi is}{N}, \ s = 1, \ldots, \frac{N-1}{2} \end{cases} \tag{28}$$

In Equation (28), the two apostrophes above the summation symbol indicate that the first and last expressions should be divided by two.

*3.2. Training Data Generation*

The pseudo-spectral method was used to generate plenty of optimal trajectories. The minimum flight time of the hypersonic vehicle was considered as the optimization target, and the generalized lift coefficient and bank angle are considered as variables to be optimized.

Optimal trajectories generated with random initial and terminal states:

Considering the varied and different flight missions of hypersonic vehicles, the information of the initial point and terminal point cannot be determined before take-off. The hypersonic vehicle needs to generate optimal controls in the light of the current mission and flight environment information; to address the problem of autonomous intelligent behavior planning of hypersonic vehicles in uncertain flight environments, it is necessary to design the trajectory generator with strong robustness to generate an optimal or sub-optimal trajectory with uncertain initial and terminal states. In this paper, a deep neural network is developed to perform as the real-time trajectory generator with high accuracy and strong stability. In this sense, a sufficient number of optimal trajectory data samples are required to train the deep neural network to predict the optimal controls. Therefore, for the training data generation, the states of the initial point and terminal point for each sample trajectory are randomly chosen in a certain range, based on which a large number of optimal trajectories are generated using the Chebyshev method.

The generation of massive optimal trajectories:

For each optimal trajectory generated by the pseudo-spectral method, we obtain the optimal discrete sequence of control and state variables with respect to discrete CGL time points. To gain more optimal state–action pairs as the training samples, random initial and terminal states are set for the Chebyshev method. On account of the inconformity of time label for each optimal trajectory, each optimal trajectory is interpolated about time.

## 4. Neural Network Design and Training

The DNN is proposed to predict the optimal trajectory control actions for a hypersonic vehicle based on its flight mission and current state. The proposed DNN is designed as a fully connected, feed-forward neural network with one input layer, multiple hidden layers, and one output layer. The neural network input consisted of six current position state quantities, six trajectory start position state quantities, and six terminal position state quantities; that is, $X_{input} = \{s_0, s_f, s_{current}\}$. The neural network output consisted of the trajectory control variables, the generalized lift coefficient and inclination angle, which is given as $X_{output} = \{\alpha, \sigma\}$.

It is worth pointing out that the input and output of the network should be normalized for effective training and fast convergence. The normalization process was as follows:

$$X_{norm} = \frac{X - X_{min}}{X_{max} - X_{min}} \tag{29}$$

where $X$ denotes the training dataset, $X_{max}$ and $X_{min}$ denote the maximum and minimum in $X$, respectively, and $X_{norm}$ is the normalized training dataset.

The activation function in the neural network model is the sigmoid function, which performs better than the ReLU function in the problem. The Adam accelerator was used for its high computational efficiency, and the loss value was calculated as the average of the expected output value and the squared sum of the errors. The loss value was the mean squared error and was calculated by:

$$loss = \frac{1}{n} \sum_{i=1}^{n} [f(x_i) - y_i]^2 \tag{30}$$

where $n$ is the total number of training samples, and $f(x_i)$ and $y_i$ are the predicted and true values, respectively.

The flowchart of the neural network training process is shown in Figure 2.

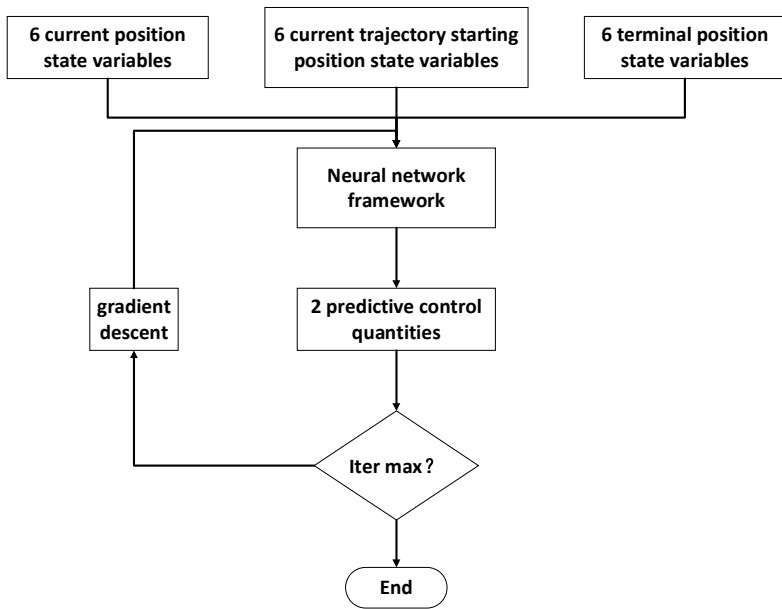

**Figure 2.** The flowchart of the neural network training process.

The pseudo-code of the Algorithm 1 used in this paper is shown below. Where $\omega$ and $\alpha$ represent the weights and bias of the neural network, $lr$ represents the learning rate of the neural network, $n\_epochs$ represents the total training batch, $batch\_size$ represents the number of samples contained in a training batch; the number of training sessions per training batch is determined by dividing the total number of training samples by $batch\_size$ rounded up. $batch\_index$ represents the index value of the training batch, the network input contains the initial value of the state $s_0$, final value of state volume $s_f$ and current state $s_{current}$, the network output includes the generalized lift coefficient $\alpha$ and inclination angle $\sigma$. $dx\_angle$ represents the range angle and subfunction environment () represents the hypersonic vehicle reentry segment model, where the input is the current control and the output is the state at the next moment.

---

**Algorithm 1** Imitation learning

---

1: Initialize network weighting values $\omega$ and $\alpha$
2: Set $lr = 0.0001, n\_epochs = 30, batch\_size = 256$
3: for $epoch = 1, n\_epochs$ do
4:　　for $batch\_index = 1, n\_batches$ do
5:　　　　obtain the optimal sequence of pseudo-spectral method ballistic $[s, a]$
6:　　　　$net\_in = \left[s_0, s_f, s_{current}\right], net\_out = [\alpha, \sigma]$ data feature extraction and normalization
7:　　　　update network parameters using Adam algorithm:
$$loss = \frac{1}{n} \sum_{i=1}^{n} \left[f(x_i) - y_i\right]^2$$
8: end for
9: Randomly generate a ballistic path by pseudo-spectral method $[s_1, a_1]$ set up data buffering $\Re$
10: *if $dx\_angle < 0.1°$* do
11:　　use neural network, input $\left[s_0, s_f, s_{current}\right]$, output $[\alpha, \sigma]$
12:　　put$[\alpha, \sigma]$ into environment(), obtain $s_{current+1}$
13:　　store samples $\left[s_0, s_f, s_{current}\right]$, $[\alpha, \ \sigma]$ to $\Re$, update $s_{current}$
14: end

---

## 5. Simulations and Result Analysis

The experiments were conducted to verify the effectiveness and generalization ability of the proposed neural network. The models of a hypersonic gliding vehicle named the high-lift common aero vehicle (CAV-H) were used to test the effectiveness of the proposed algorithm. The mass of CAV-H was 907 kg, and its aero reference area was 0.4839 m$^2$. The CAV-H had a high maximum lift-to-drag ratio of *E\* = 3.24*, and the corresponding lift coefficient $C_L^*$ was 0.45. The pneumatic reference area was s$_{\text{ref}}$ = 0.8. The gravitational acceleration was *$g_0$ = 9.8* m/s$^2$, and the Earth radius was considered to be $R_0$ = 6378 km.

The parameters of the starting and terminal points of the glide section of a hypersonic vehicle are given in Table 1. The constraints that the ballistic optimization needs to meet are listed in Table 2.

**Table 1.** Initial and termination conditions.

| Parameter | Value Range |
|---|---|
| Initial height $h_0$ | 41 km~46 km |
| Initial longitude $\theta_0$ | −2°~2° |
| Initial latitude $\varphi_0$ | −2°~2° |
| Initial velocity $V_0$ | 5300 m/s |
| Initial track angle $\gamma_0$ | 0° |
| Initial course angle $\psi_0$ | 90° |
| Final longitude $\theta_f$ | 38°~42° |
| Final latitude $\varphi_f$ | 18°~22° |

**Table 2.** Process constraints of trajectory planning.

| Parameter | $\dot{Q}(\frac{kW}{m^2})_{max}$ | $\bar{q}(kPa)_{max}$ | $n(g_0)_{max}$ | Generalized Lift Coefficient | Heeling Angle (°) |
|---|---|---|---|---|---|
| Value | 2000 | 500 | 3 | $0 \leq \lambda \leq 2$ | $-80 \leq \sigma \leq 80$ |

In Table 2, $\dot{Q}_{max}$ denotes maximum heat flow density, $\bar{q}_{max}$ represents the maximum dynamic pressure, and $n_{max}$ is the maximum normal overload.

### 5.1. Generation of the Training Data

The Chebyshev pseudo-spectral method was used to generate 5000 trajectories, and the serial variations geocentric distance, longitude, latitude, velocity, control volume, generalized lift coefficients, and inclination angles are shown in Figures 3 and 4. The ballistic data were interpolated to obtain the ballistic states at 1-s intervals, and the 5000 trajectory data samples were summed to form a total data sample. The sample size was approximately 7.5 million ballistic data states.

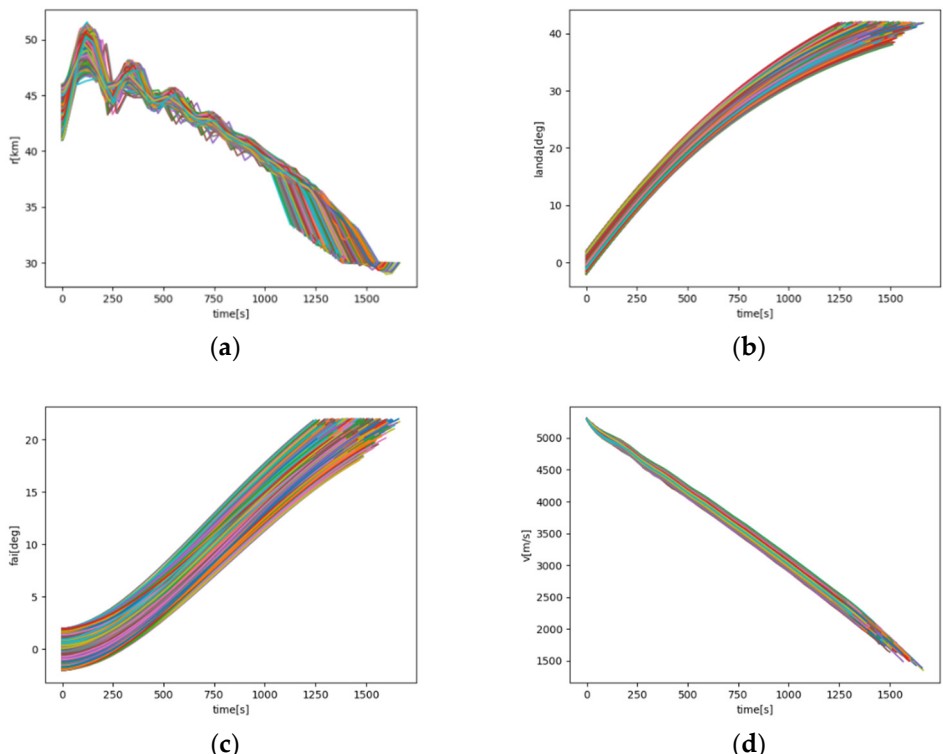

**Figure 3.** State of training data. (**a**) Height–time curve; (**b**) Longitude–time curve; (**c**) Latitude–time curve; (**d**) velocity–time curve.

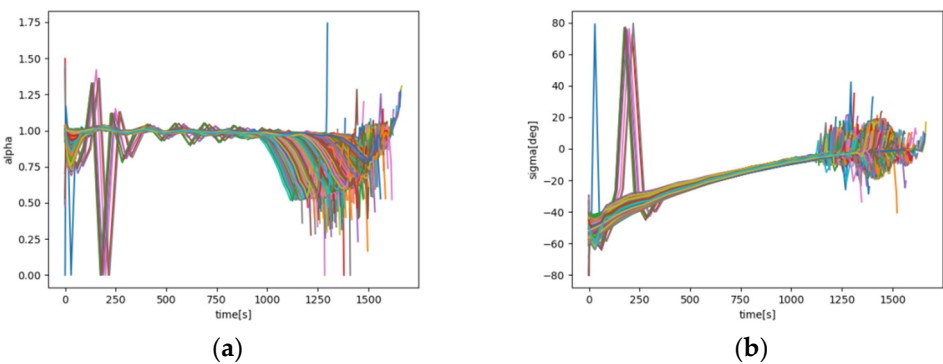

**Figure 4.** Control of training data. (**a**) Generalized lift coefficient–time curve; (**b**) Heeling Angle–time curve.

*5.2. Training Process of the DNN*

The loss value for 10,000 training epochs is shown in Figure 5. When the neural network was trained using the sigmoid activation function, the loss value could converge quickly and converge in 0.001. The data of 5000 trajectories were divided into a training set consisting of 4000 trajectories and a test set consisting of 1000 trajectories. In addition, the sigmoid and ReLU activation functions were used for comparison.

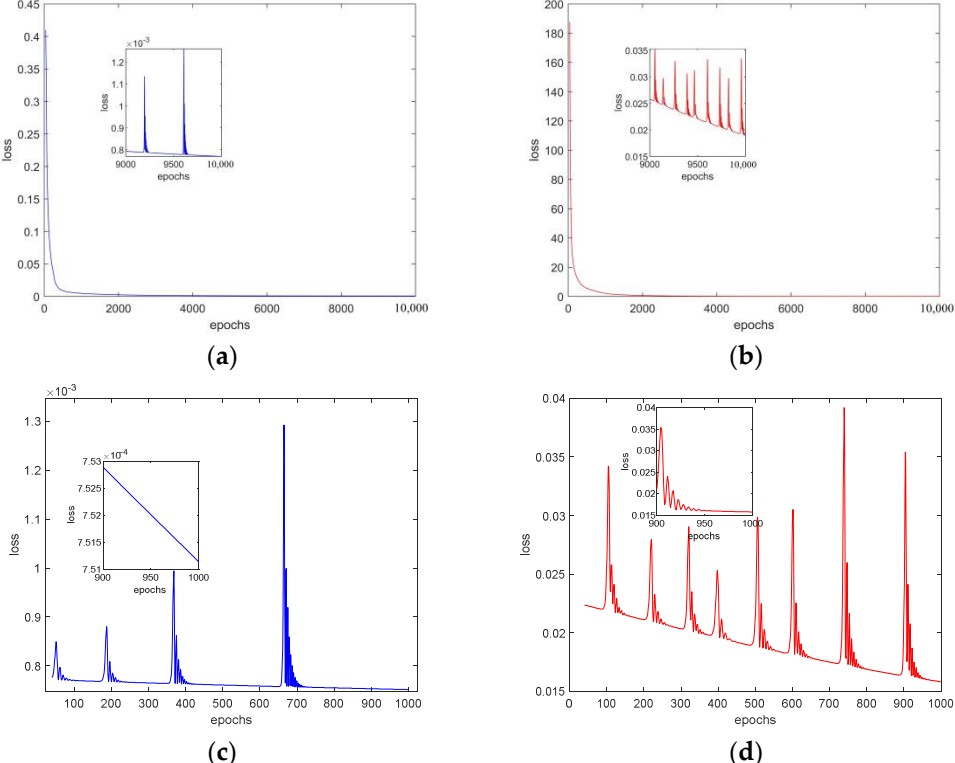

**Figure 5.** Training results of deep neural network. (**a**) The training loss for sigmoid activation function epochs; (**b**) the training loss for the ReLU activation function; (**c**) the test loss for the sigmoid activation function; (**d**) the test loss for the ReLU activation function.

The loss values for the ReLU and sigmoid activation functions are shown in Figure 5, respectively. It can be seen that the loss values were larger on the testing set, but the overall loss value was stable and at a relatively low level. The results showed that the loss value on the test set for the sigmoid function was near 0.001, while that of the ReLU function was above 0.05. Thus, the sigmoid activation function made the loss function converge to a smaller value, which is chosen as the activation function for the network.

*5.3. Random Single Trajectory Error Analysis*

In the simulations, the initial and terminal states of the trajectory are randomly generated in a certain range, and the state sequence is used as the network input. The trained deep neural network is used to predict the values of the trajectory control variables (generalized lift coefficient and inclination angle), and the predicted values are compared to the expected values that were obtained by the pseudo-spectral method to verify the effectiveness of the neural network. The comparison results of the predicted and expected output values are shown in Figures 6 and 7, where it can be seen that the predicted and expected output values coincided well during the whole flight, and the error is basically under 0.02, which verified the deep neural network's capability in online planning and the prediction of the generalized lift coefficient and inclination angle values.

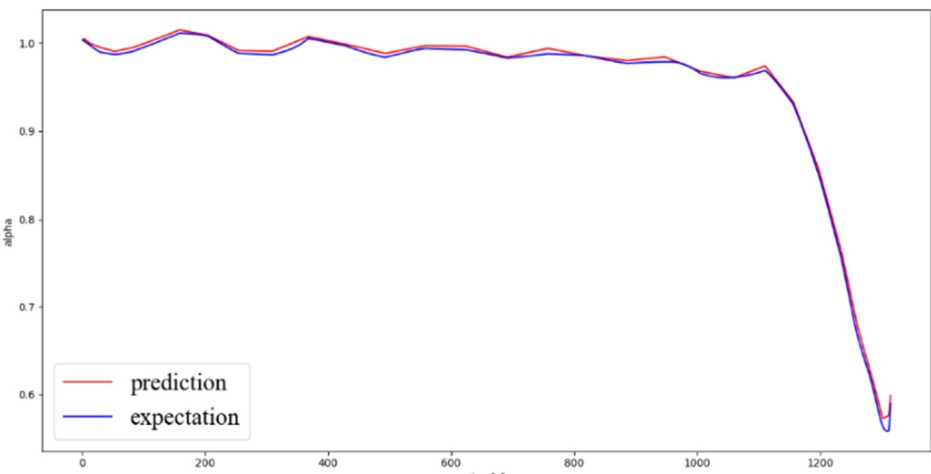

**Figure 6.** Comparison of the predicted and expected values of the generalized coefficient of the lift.

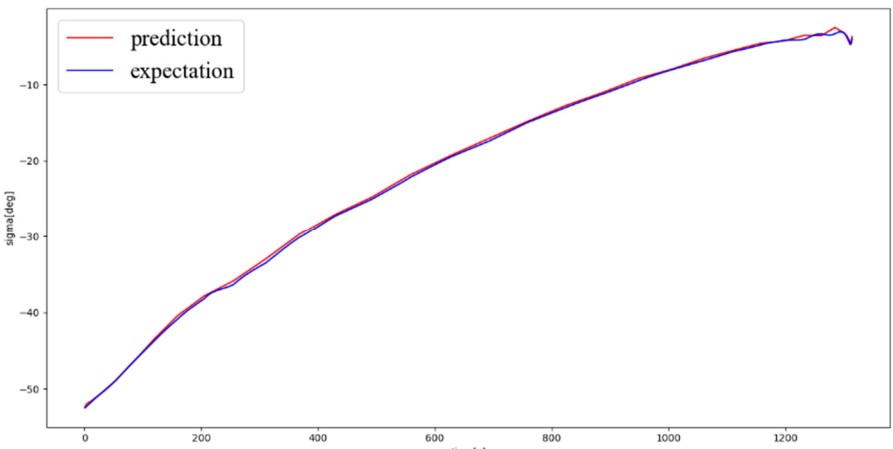

**Figure 7.** Comparison of the predicted and expected values of the inclination angle.

*5.4. Validation with Vehicle Dynamics Model*

The three-DOF model of the hypersonic vehicle reentry phase was used to further verify the prediction performance of the proposed model. The neural network consisted of eight layers, each of which had 500 neurons. There is a total of 40 batches in training, and the number of samples per batch was set to 256. A single trajectory is taken as an example, and a random trajectory was generated by the pseudo-spectral method. The start and end position conditions set by the pseudo-spectral method are substituted into the trained neural network for testing, and the comparison of the flight paths estimated by the pseudo-spectral method and those predicted by the neural network is used to analyze the output error of the neural network model. The generalized lift coefficient and inclination angle are presented in Figures 8 and 9, respectively, where it can be seen that the predicted and estimated values coincided well. The error curves of the generalized lift coefficient and inclination angle are presented in Figures 10 and 11. Based on the results, the error of the generalized lift was within $\pm 0.01°$, and the error of the inclination angle was within $\pm 0.02°$. The numerical values of the errors of the neural network prediction are given in Table 3. As shown in Table 4, the geocentric distance error was within 1 km, the longitude and latitude errors were 0.1° and 0.03°, respectively, and the velocity error was 4 m/s.

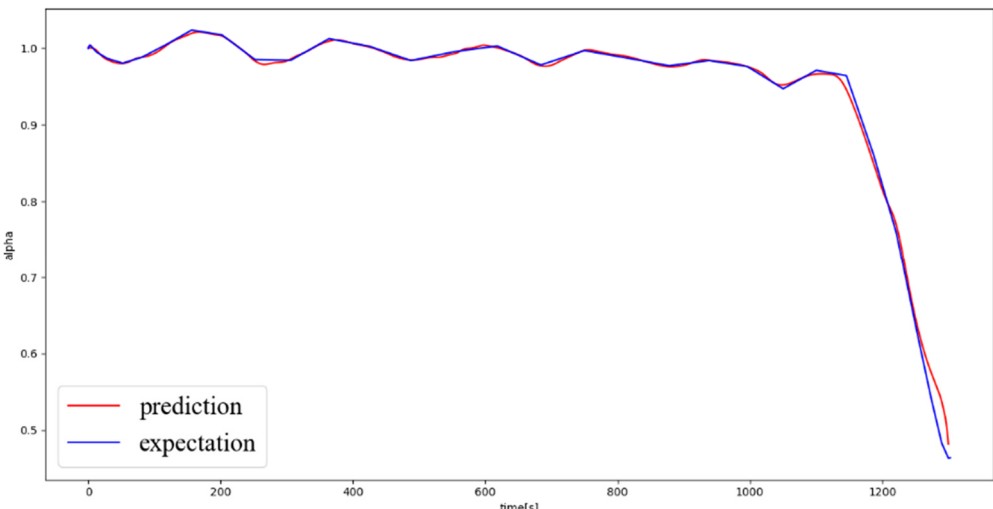

**Figure 8.** Comparison of the predicted and expected values of the generalized coefficient of the lift.

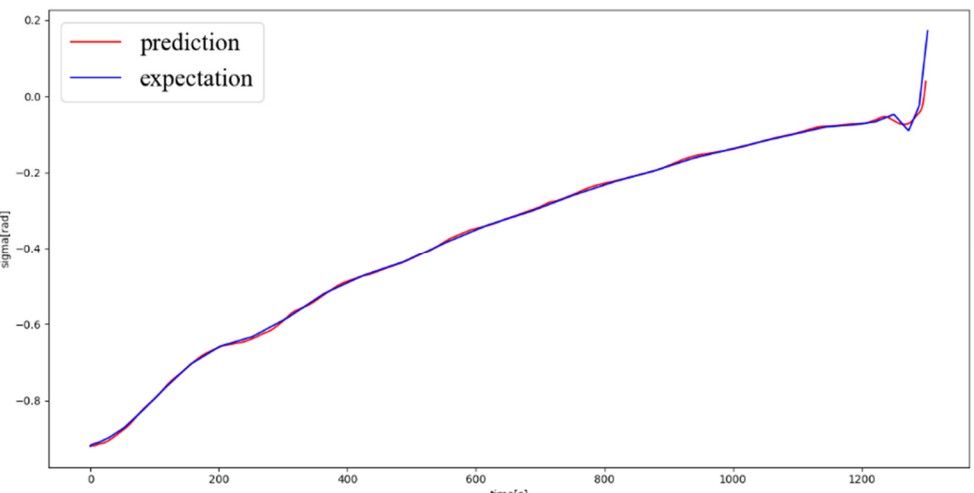

**Figure 9.** Comparison of the predicted and expected values of the inclination angle.

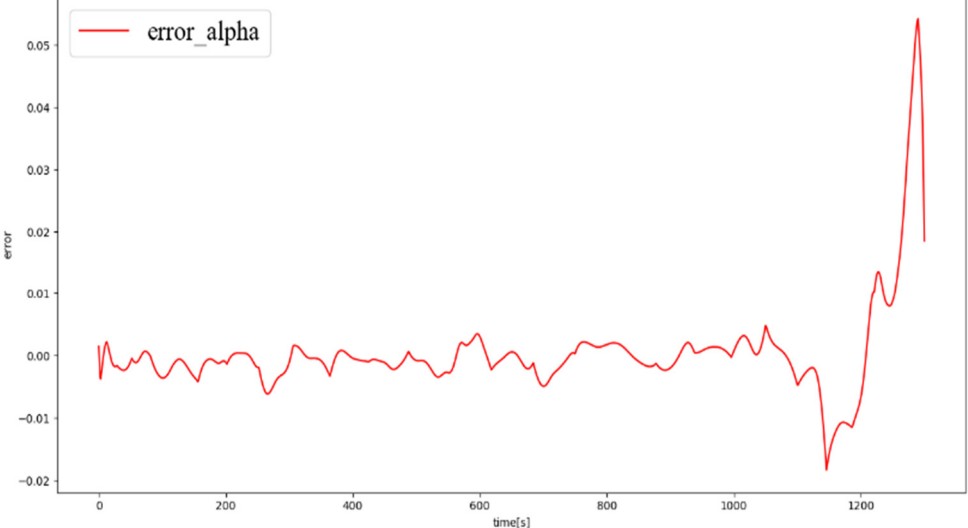

**Figure 10.** Generalized lift error change with time.

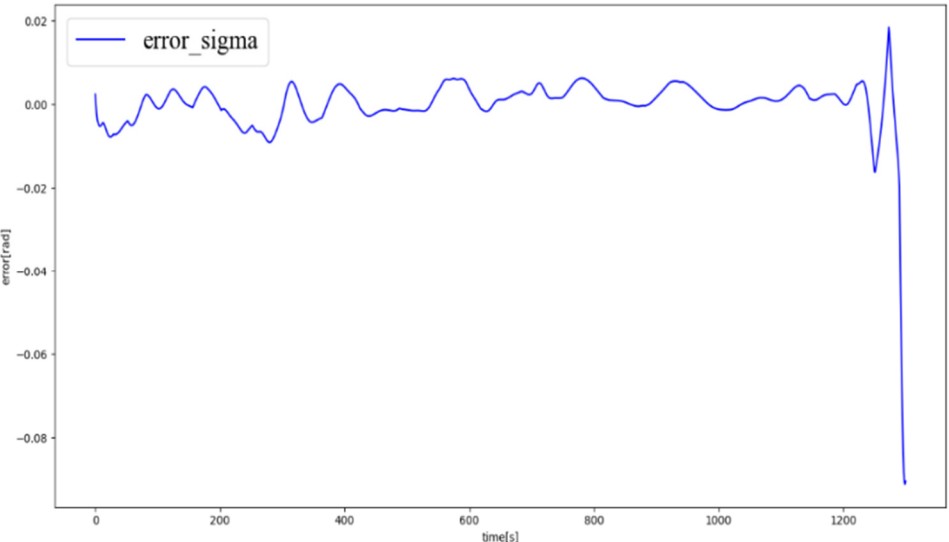

**Figure 11.** Inclination angle error change with time.

**Table 3.** Error statistics.

|  | **Actual Vehicle Position** | **Predicted Vehicle Position** | **Position Error** |
|---|---|---|---|
| Altitude (m) | 30,151 | 30,940 | 789 |
| Longitude (°) | 34.84 | 34.74 | 0.10 |
| Latitude (°) | 18.16 | 18.19 | 0.03 |
| Velocity (m/s) | 2267 | 2271 | 4 |

**Table 4.** Error statistics of Monte Carlo simulation (90 percent probability).

|  | **The Absolute Terminal Longitude** | **The Absolute Terminal Latitude** | **The Absolute Terminal Range Angle** |
|---|---|---|---|
| error (°) | 0.042 | 0.125 | 0.126 |

*5.5. Monte Carlo Simulation Verification*

In order to demonstrate the generalization ability of the developed neural network model, the Monte Carlo ballistic simulation and error analysis were carried out. In the simulations, random ballistic beginning and end state parameters were used, and there were 1000 target trajectories. The Monte Carlo simulation was performed using an online planning method based on the neural network.

The analysis results are shown in the following table.

## 6. Conclusions

In this study, a deep neural network-based method is developed to achieve fast prediction of optimal trajectories for a hypersonic vehicle. First, the reentry phase of a hypersonic vehicle is formulated as an optimal control problem, and the pseudo-spectral method is developed to provide optimal solutions for DNN training. The developed DNN model is optimized on the test set regarding the numbers of layers and neurons, learning rate, and activation functions. Based on the optimized DNN model, the DNN-based method and improvement techniques are developed and employed to solve the optimal trajectory problem. The proposed method is verified by numerical simulations, and the results demonstrate that the DNN-based method has the advantages of fast solving speed and excellent convergence.

The proposed method provides an original idea for the online trajectory optimization of a hypersonic vehicle, and the trajectory optimization of the entire trajectory can be accomplished accurately in only a few seconds. Similarly, the proposed method can be applied to other models in the aerospace field, such as lunar landing and asteroid detection models. In future work, more complex flight missions and more rigorous constraints, including no-fly zones, are considered to verify the effectiveness of the proposed method. We will also adopt more elaborate network structures to enhance the learning accuracy.

**Author Contributions:** Formal analysis, M.Y.; funding acquisition, M.L.; investigation, J.W.; supervision, H.L.; visualization, Y.W.; writing—original draft preparation, Y.W.; writing—review and editing, J.W. All authors have read and agreed to the published version of the manuscript.

**Funding:** This research was funded by the National Natural Science Foundation of China, Grant No. 62103452 and No. 62003375.

**Institutional Review Board Statement:** Not applicable.

**Informed Consent Statement:** Not applicable.

**Data Availability Statement:** All data used during the study appear in the submitted article.

**Acknowledgments:** The study described in the paper was supported by the National Natural Science Foundation of China (Grant NO.62103452, NO.62003375). The authors fully appreciate their financial support.

**Conflicts of Interest:** The authors certify that there are no conflict of interest with any individual/organization for the present work.

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
