# Peer review of "A Real-Time Trajectory Optimization Method for Hypersonic Vehicles Based on a Deep Neural Network"

_aerospace, doi:10.3390/aerospace9040188_

Round 1

Reviewer 1 Report

In this article, a deep neural network is developed to solve the optimal trajectory generation problem in real time for hypersonic vehicles.
In general, manuscript is interesting and easy to understand. However, there are parts which should be revised:

1.    Avoid expressions like: “hottest technology areas”.
2.    The manuscript contains some errors in grammar which need to be corrected.
3.    Some abbreviations and acronyms are not defined (e.g. DNN, DOF).
4.    Line 138, Earth (uppercase).
5.    Standardize mathematical symbols. Different symbols being used to express the same variable (e.g. velocity). In addition, there are a lot of variables that have not been defined throughout the text (e.g. Ro, go, Vc, rho...).
6.    Line 144. The authors say that the variables were substituted in Eqs. 8 – 13. But aren't such equations the result of the substitution? It was not clear.
7.    Citation of figure 1 in the text is incorrect.
8.    Error in the equation 29.
9.    Line 304 - Change terms to: ReLU e Adam.
10.    Explain in more detail the results shown in graphs 12-15.
11.    Reference 29 appears to be a base work for the manuscript. In the contributions paragraph, the authors should more explicitly present the differences in relation to 29, which I believe is related to the 3D trajectory. This information does not appear in the paragraph on line 107.

Reviewer 2 Report

The paper presents a DNN-based trajectory planning and control method for   hypersonic vehicles.

the paper is well written and technically sound. I just have some minor comments/doubts.

Regarding the network, I don't understand why the authors give “trajectory starting position state” as input to the neural network. The current position state and the terminal position state should be enough to compute the optimal trajectory and control. Formulating the problem would be a bit more difficult but in the end I think you would need much less training data/time.

line 155, define sigma and alpha.

line 193, fig?.

line 195, action symbol alpha is the same as control vector alpha.

section 5, fix the symbols.

Figures, put variable symbols for y-axis

line 407,409, figure numbers incorrect.

figure, 14,15,16 could be replaced by a table.

Round 2

Reviewer 1 Report

Manuscript was significantly improved.